# Improving access to health services through health reform in Lesotho: Progress made towards achieving Universal Health Coverage

**Melino Ndayizigiye**[1], **Lao-Tzu Allan-Blitz**[2,3]*, **Emily Dally**[4], **Seyfu Abebe**[1], **Afom Andom**[4], **Retsepile Tlali**[1], **Emily Gingras**[1], **Mathabang Mokoena**[1], **Meba Msuya**[4], **Patrick Nkundanyirazo**[4], **Thiane Mohlouoa**[1], **Fusi Mosebo**[1], **Sophie Motsamai**[1], **Joalane Mabathoana**[1], **Palesa Chetane**[1], **Likhapha Ntlamelle**[1], **Joel Curtain**[4], **Collin Whelley**[5], **Ermyas Birru**[6], **Ryan McBain**[7], **Di Miceli Andrea**[8], **Dan Schwarz**[2,9], **Joia S. Mukherjee**[2,4]

1 Partners In Health, Maseru, Lesotho, 2 Division of Global Health Equity: Department of Medicine, Brigham and Women's Hospital, Boston, Massachusetts, United States of America, 3 Department of Pediatrics, Boston Children's Hospital, Boston, Massachusetts, United States of America, 4 Partners In Health, Boston, Massachusetts, United States of America, 5 Homebase Center for Common Concerns, San Francisco, California, United States of America, 6 University of Washington, Seattle, Washington, United States of America, 7 RAND Corporation, Boston, Massachusetts, United States of America, 8 Analysis Group Inc., Boston, Massachusetts, United States of America, 9 Ariadne Labs, Boston, Massachusetts, United States of America

☯ These authors contributed equally to this work.
* Lallan-blitz@partners.org

## Abstract

In 2014 the Kingdom of Lesotho, in conjunction with Partners In Health, launched a National Health Reform with three components: 1) improved supply-side inputs based on disease burden in the catchment area of each of 70 public primary care clinics, 2) decentralization of health managerial capacity to the district level, and 3) demand-side interventions including paid village health workers. We assessed changes in the quarterly average of quality metrics from pre-National Health Reform in 2013 to 2017, which included number of women attending their first antenatal care visit, number of post-natal care visits attended, number of children fully immunized at one year of age, number of HIV tests performed, number of HIV infection cases diagnosed, and the availability of essential health commodities. The number of health centers adequately equipped to provide a facility-based delivery increased from 3% to 95% with an associated increase in facility-based deliveries from 2% to 33%. The number of women attending their first antenatal and postnatal care visits rose from 1,877 to 2,729, and 1,908 to 2,241, respectively. The number of children fully immunized at one year of life increased from 191 to 294. The number of HIV tests performed increased from 5,163 to 12,210, with the proportion of patients living with HIV lost to follow-up falling from 27% to 22%. By the end of the observation period, the availability of essential health commodities increased to 90% or above. Four years after implementation of the National Health Reform, we observed increases in antenatal and post-natal care, and facility-based deliveries, as well as child immunization, and HIV testing and retention in care. Improved access to and

**Data Availability Statement:** All relevant data are within the paper and its Supporting information files.

**Funding:** Funding for this study came from the Skoll Foundation (grant number 19-44478) (JM) and the Wagner Foundation (project code AVA0436) (JM). The funders had no role in study design, data collection and analysis, decision to publish, or preparation of the manuscript.

**Competing interests:** The authors have nothing to declared no competing interests exist.

utilization of primary care services are important steps toward improving health outcomes, but additional longitudinal follow-up of the reform districts will be needed.

## Introduction

According to the World Bank report in 2017, the Kingdom of Lesotho had some of the poorest health outcomes in the world with some of the highest rates of both human immunodeficiency virus (HIV) infection and tuberculosis [1]. Over the last two decades, Lesotho received a significant amount of international aid primarily focused on controlling HIV and tuberculosis. Yet unlike other African countries, Lesotho did not use those funds to strengthen primary healthcare. Thus, at the end of the Millennium Development Goals era in 2015, Lesotho had persistently high rates of infant and maternal mortality [2–4] as well as child malnutrition [5, 6]. Disease-focused programs, largely managed by international organizations, have struggled to meet targets for case-finding and treatment of HIV and Tuberculosis [7]. In 2017, there were over 13,000 new HIV infections among children due to gaps in prevention of vertical transmission [1, 8]. The poor performance of those programs led to a widespread recognition of the need for horizontal, multifaceted health-system strengthening to address all causes of mortality [9, 10]. Further, healthcare financing in Lesotho is limited; the government of Lesotho spends more than 10% of their national budget on health, yet costs of the country's central hospital consumes more than half of the public expenditures on health [11].

In 2006, the Ministry of Health and Social Welfare (MOHSW), the Clinton Health Access Initiative, and Partners In Health (PIH) implemented a multipronged intervention and demonstrated improved access to care and healthcare utilization among pregnant, laboring and post-partum women [12, 13]. Based on that success, the MOHSW solicited the support of PIH in the design, implementation, and evaluation a national health reform in primary care clinics [14–16]. Four districts were selected by the MOHSW to pilot the National Health Reform interventions: Berea, Leribe, Butha-Buthe and Mohale's Hoek, which spanned an estimated catchment population of 815,520 individuals (40% of the country's total population). All health facilities in those districts were included. Through the baseline auditing of the 70 facilities [17] three primary inter-related deficits were identified: 1) insufficient staffing, inadequate space, and a lack of supplies to meet the burden of disease for the catchment areas of the clinics, 2) lack of district-level managerial capacity, and 3) a Village Health Program untethered from the health system and depended on undercompensated, unsupervised labor of village health workers utilizing a variety of one-off interventions [18].

Subsequently, MOHSW and PIH implemented a multi-tiered national health reform within those four districts. An interim report in 2019 of outcomes from that reform revealed increased antenatal care visits, facility-based deliveries, and improved inter-facility transfer of complex cases (S1 Text). The present report aims to evaluate the impact of the numerous interventions of the national health reform on various outcome metrics.

## Methods

### Ethics statement

The Harvard Human Research Protection Program granted ethical approval for this study (IRB17-19888), as did the Lesotho National Health Research Ethics Committee (id 117–2017). Informed consent was deemed not necessary as the reform interventions were done on a systems level.

### Overview of national health reform interventions

The National Health Reform was implemented in 2014 (see S2 Text for specific interventions and S3 Text for the detailed study protocol) aiming to address the three key deficiencies: service delivery, district managerial capacity, and the Village Health Program. Regarding service delivery, at the facility level, the DHS data for each district was compared with the catchment area to set targets for service delivery using the PIH Universal Health Coverage planning tool [19, 20]. Based on that assessment inputs to the system were aligned with the expected need. Specific interventions included supplies of required equipment for facility-based deliveries and hiring of 24-hour midwife coverage, the establishment of maternal waiting homes, provision of heating supplies and food. To improve supply chain management and reduce stock outs of essential medications, we used morbidity-based mapping and tracer commodities in combination with the introduction of 13 additional pharmacists. In order to decentralize district managerial capacity, District Health Management Team were trained and implemented within each district, which were responsible for overseeing health services delivery. Finally, 5,359 village health workers were recruited to increase coverage in all villages within the catchment areas, and trained via a standardized PIH community health worker curriculum [21].

Over the period of the intervention, there were several deviations from the protocol (S3 Text) both from an implementation standpoint as well as from an analysis standpoint. First, in addition to the four pilot villages, the health reform interventions were initially planned for implementation in the central ministry of health. Frequent internal turnover of government staff precluded that implementation. From an analysis standpoint, we were unable to collect the population level metrics on tuberculosis outcomes, thus those outcomes were excluded from the present report. Similarly, the data collected regarding initiation of antiretroviral therapy among patients diagnosed with HIV contained numerous missing entries, thus was excluded from the analysis to avoid introducing bias.

### Data collection

Baseline data collection was done in all the health facilities in the four districts from July 2013 to June 2014 via standardized collection tools. Those tools included standardized interviews with key stakeholders, as well as population-level assessment of key facility health metrics that were used as our outcomes of interest as described in detail below (S4 Text). Those data were collected by a monitoring and evaluation team from the PIH and MOHSW who went to all facilities and collected data from the facility registers over a period of six months. The collection of the data was done on password protected and encrypted electronic tablets.

### Study design and data analysis

The primary outcomes of interest were defined in three domains: 1) utilization of healthcare services, 2) children receiving immunizations at the facility, and 3) HIV testing and retention in care. Those domains were selected as they were most readily accessible for analysis, and most likely directly impacted by the specific interventions (S1 Text). With regards to maternal health, the outcome measures were the change pre- and post-intervention in average quarterly number of women attending their first and fourth antenatal care visits, average quarterly number of women attending their post-natal care visits, average quarterly number of health centers adequately equipped to provide a facility-based birth, average quarterly number of facility-based births that occurred. With regards to childhood immunizations, we evaluated the change in the average quarterly number of children who were immunized at one year of life at the clinic. Complete immunization at one year of life was defined as having received BCG, Hepatitis B, Polio, Diphtheria-Tetanus-Pertussis, *Haemophilus influenzae* type B, Pneumococcal

**Table 1. Nationwide annual demographic trends across Lesotho at the population level, 2011–2019.**

| Year | 2011 | 2012 | 2013 | 2014* | 2015 | 2016 | 2017 | 2018 | 2019 |
|---|---|---|---|---|---|---|---|---|---|
| Crude Birth Rate (per 1,000 people)[†] | 28.8 | 28.6 | 28.3 | 28.1 | 27.8 | 27.5 | 27.2 | 26.8 | 26.4 |
| Incidence of HIV, ages 15–49 (per 1,000 uninfected pop. ages 15–49)[††] | 21.4 | 21.0 | 20.3 | 18.8 | 17.1 | 15.1 | 12.7 | 10.7 | 9.8 |
| Rural population (% of total population)[§] | 74.7 | 74.3 | 73.9 | 73.5 | 73.1 | 72.7 | 72.3 | 71.8 | 71.4 |
| Population, female (% of total population)** | 51.3 | 51.2 | 51.1 | 50.9 | 50.9 | 50.8 | 50.7 | 50.7 | 50.7 |
| Annual health expenditure per capita (current US$)[§§] | 125.7 | 117.3 | 115.8 | 113.3 | 104.2 | 89.5 | 110.9 | 124.7 | 124.2 |
| Gross Domestic Product per capita (current USD)[π] | 1,287 | 1,230 | 1,167 | 1,195 | 1,146 | 1,019 | 1,103 | 1,192 | 1,113 |

*The National Health Reformation started on Q2 2014

[†] The crude birth rate indicates the number of live births occurring during the year, per 1,000 population estimated at midyear.

[††] The incidence of HIV is computed as the number of new HIV infections among uninfected populations ages 15–49 expressed per 1,000 uninfected population in the year before the period.

[§] Rural population refers to people living in rural areas as defined by national statistical offices. It is calculated as the difference between total population and urban population.

** The proportion female in the population is based on the population count, which counts all residents regardless of legal status or citizenship

[§§] Estimates of annual health expenditures include healthcare goods and services consumed during each year. Data are in current U.S. dollars.

[π] Gross domestic product per capita is equal to the gross domestic product divided by midyear population. Data are in current U.S. dollars.

The table shows national population level trends in birth rate, incidence of HIV, proportion of the total population living in rural areas, proportion of the total population identifying as female, annual health expenditure per capita, and gross domestic product.

(conjugate), Rotavirus, Measles-Mumps-Rubella, and Human Papilloma Virus [22]. Finally, with regards to HIV testing and retention in care, our primary outcomes of interest were the proportion of patients living with HIV and enrolled in care who were lost to follow-up, the average quarterly number of HIV tests performed at health facilities, and the average quarterly number of new HIV cases detected.

Our secondary outcomes of interest were average quarterly number of maternal emergency referrals placed and transported to the health facility and the availability of essential medications. Those two metrics were selected as there was no emergency referral system prior to the national health reform, and the availability of essential medications was subject to numerous confounding variables that we were unable to control for, thus it was selected as a secondary outcome.

In order to assess for potential confounding by differences in the population before and after the National Health Reform, evaluated the change population demographic variables over the duration of the study (Table 1) [23]. We then conducted a one arm pre- and post-intervention time series analysis, evaluating changes in the outcomes above after three years of operationalizing the National Health Reform compared to before (2013). Data were collected in 2017, thus that was considered the end point for the study, however the activities of the National Health Reform are ongoing. Data were analyzed using STATA software version 15.1 and interrupted time series were performed to assess the trends on different indicators. Results are reported in accordance with STROBE guidelines (S1 Checklist). Data are available as a supplemental file (S1 Data).

## Results

### Maternal health

The quarterly average number of women who attended their first antenatal care visit rose from 1,877 (SD +/- 263) pre-National Health Reform to 2,729 (SD +/- 287) in the last year of the intervention, with similar though smaller trends noted for antenatal care visits (see Fig 1a).

a: Quarterly average number of women attending prenatal care visits pre-and post-National Health Reformation

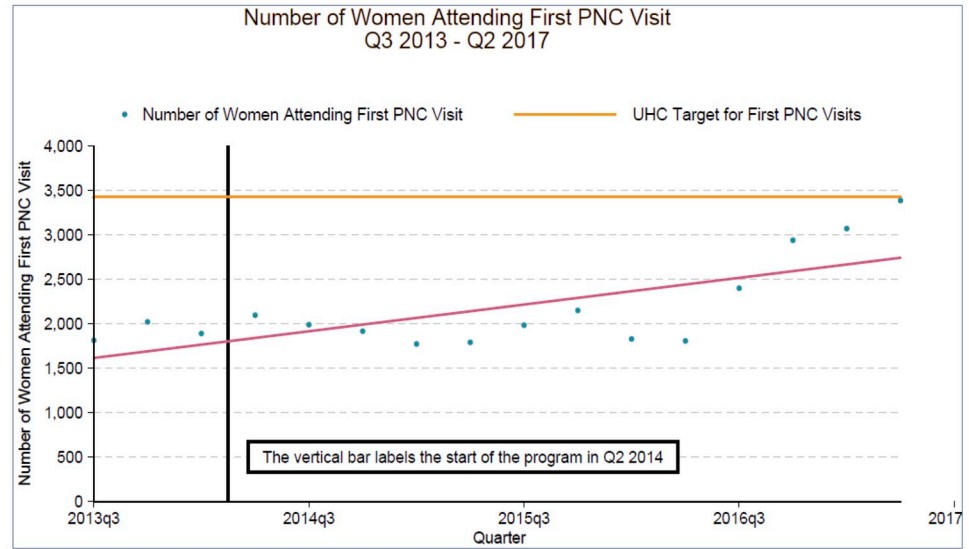

b: Quarterly average number of women attending their first and fourth antenatal care visits pre-and post-National Health Reformation

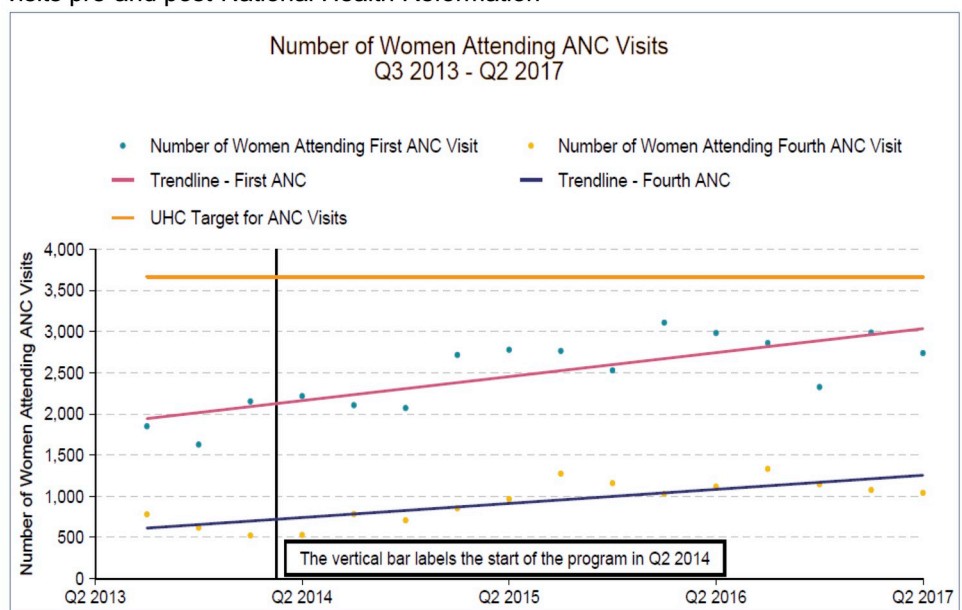

**Fig 1. Changes in quarterly average number of women attending prenatal and antenatal care visit Q3 2013 –Q2 2017.** a. shows the changes in quarterly average number of women attending prenatal care visits in four rural districts across Lesotho. b shows the changes in quarterly average number of women presenting for their first and fourth antenatal care visits in four rural districts across Lesotho. Both figures show the change over time beginning in the third quarter of 2013 (before the National Health Reform) through the second quarter of 2017 (nearly three years into the National Health Reform).

Similarly, the percent of visits at local health centers used for fourth antenatal care visit rose from 69% to 78% ($p$-value $< 0.01$). Importantly, the number of health centers adequately equipped to provide a facility-based delivery increased from 3% to 95% with an associated increase in deliveries occurring at a health facility from 2% to 33% (see Fig 2).

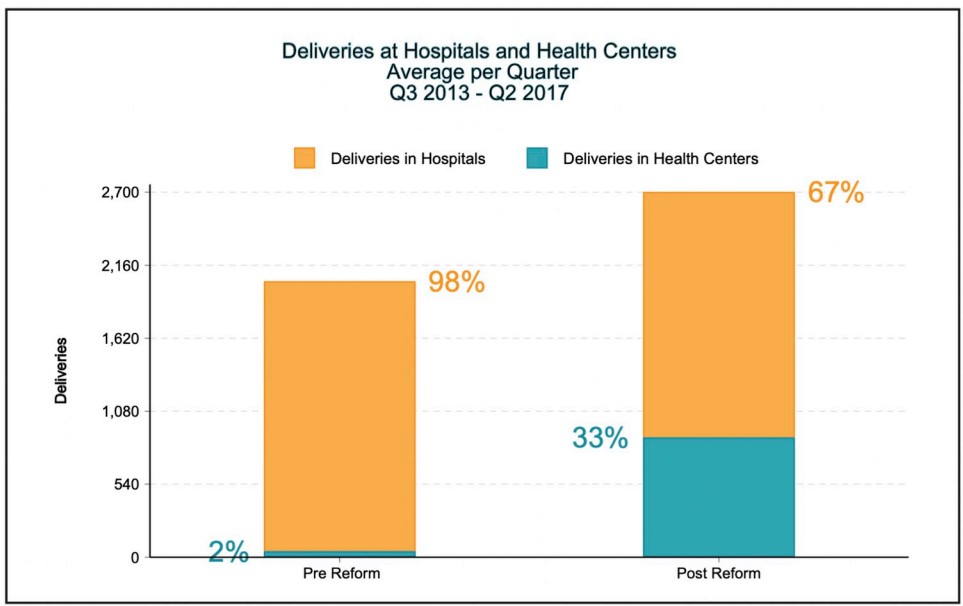

**Fig 2. Percent of deliveries occurring in a hospital or health center in Lesotho.** Fig 2 shows the percentage of births occurring within health centers in four rural districts across Lesotho, comparing pre- and post-National Health Reform interventions.

Emergency referral and transportation of obstetric complications increased from 2.3 transports pre-intervention to 42 total transportations post-intervention, with similar increases noted across all districts. Additionally, by the end of the National Health Reform, the availability of essential medical supplies, tracer drugs, antiretroviral therapies, and vaccines to 90% or above. Notably, no quantitative data had been collected about the availability of such supplies prior to the National Health Reform, thus systems were consequently implemented to track availability of all essential health commodities within each district. Overall, the number of postnatal care visits increased from a quarterly average of 1,908 (SD +/- 105) pre-National Health Reform to 2,241 (SD +/- 545) (see Fig 1b).

## Childhood immunizations at one year of life

The number of children fully immunized at one year of life increased from a quarterly average of 191 (SD +/- 108) to 294 (SD +/- 238) (*p*-value 0.148) (see Fig 3). Testing for HIV among children (from 6 weeks to 18 months old) rose from a quarterly average of 301 (SD +/- 147) to 456 (SD +/- 262).

## HIV prevention and care continuum

The proportion of patients living with HIV enrolled in care receiving antiretroviral therapy who were lost to follow-up pre-National Health Reform fell from 27% across all health facilities to 22% post-National Health Reform (*p*-value<0.05). Further, the number of HIV tests performed at health facilities increased from a quarterly average of 5,163 (SD +/- 2,990) pre-National Health Reform to 12,210 (SD +/- 6,915) (Fig 4). The quarterly average cases detected increased from 592 (SD +/- 301) to 735 (SD +/- 587). Among those, the number of infected pregnant women increased from a quarterly average of 172 (SD +/- 74) to 199 (SD +/-). The increase in testing resulted in an increased case detection rate from a quarterly average 6.3

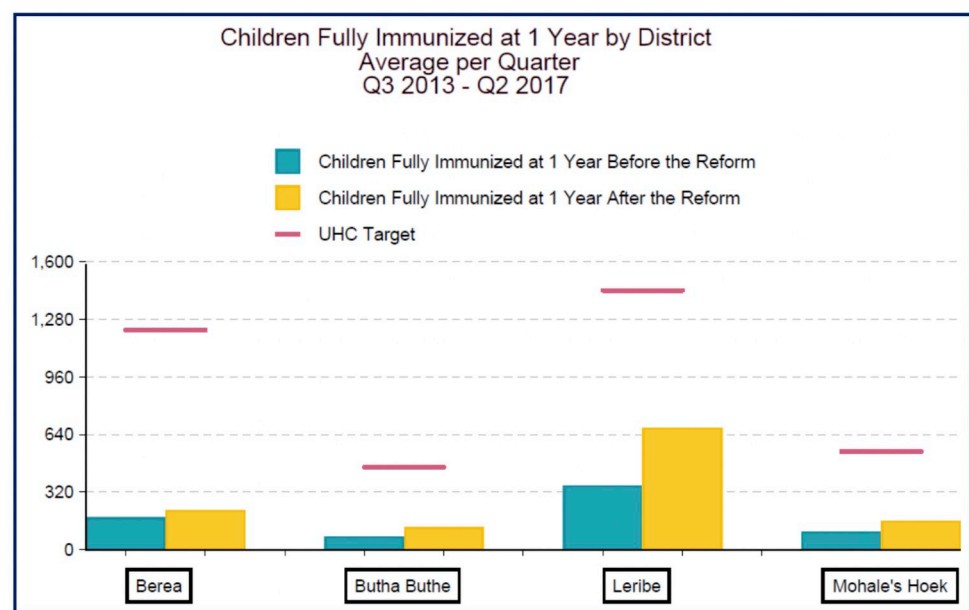

**Fig 3. Quarterly average number of children fully immunized at one year of life by district in Lesotho, Q3 2013 – Q2 2017.** Fig 3 shows the quarterly average number of children fully immunized at one year in each of four rural districts across Lesotho in which National Health Reform interventions took place, comparing pre- and post-intervention results. Notes: (Figure 19) [1] The pink line represents the line of best fit. The slope is 71.940 [2] Excludes hospitals.

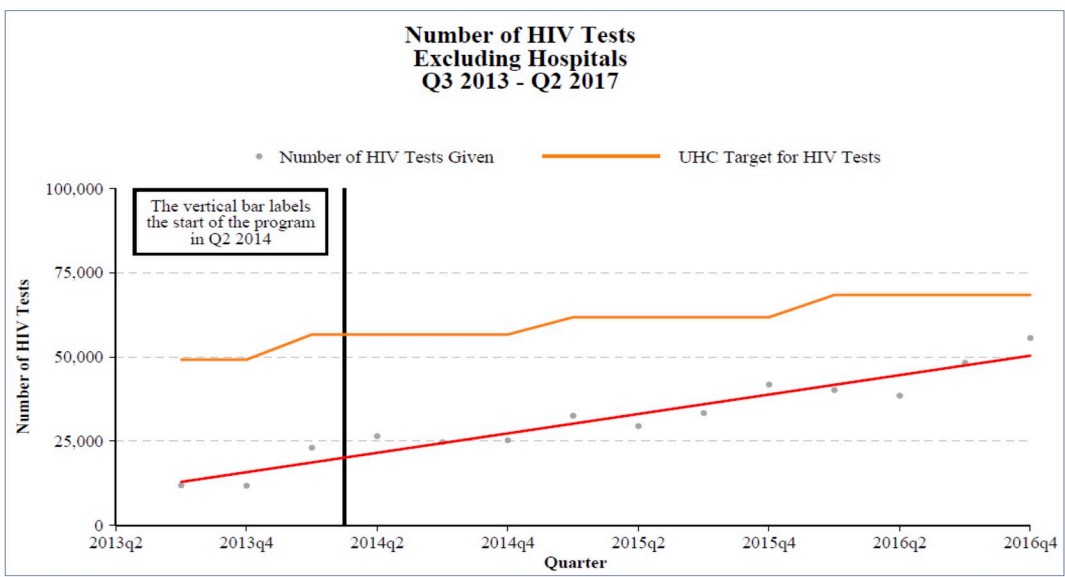

**Fig 4. Increased number of HIV tests at health facilities Q3 2013 –Q2 2017.** Fig 4 shows the trend in the number of HIV tests performed outside of the hospital across four rural districts in Lesotho over time, plotted against the targets for HIV testing established by the Universal Health Coverage goals. The figure shows the change over time beginning in the third quarter of 2013 (before the National Health Reform) through the fourth quarter of 2016 (more than two years into the National Health Reform).

cases per quarter (SD +/- 4.6) to 7.2 (SD +/- 5.5) before and after the National Health Reform, respectively.

## Discussion

We report on the extensive National Health Reform undertaken in Lesotho with the support of PIH, as well as various improvements in health outcomes observed after the reform period. The three main interventions of the National Health Reform were supply-side system strengthening, improving district managerial capacity, and re-vitalizing the village health worker program. After implementation of those interventions, though no comment on causation can be made, we observed improvements in maternal health outcomes, the proportion of children fully immunized by one year of age, and in HIV diagnostic capacity as well as retention in care.

The improvements in maternal and child health metrics observed included access to care and care utilization such as increased in antenatal and postnatal care visits and facility-based deliveries. Additionally, emergent referrals from health centers to hospitals are now possible with the establishment of emergency referral systems. Access to antenatal care and facility-based deliveries both have been shown to improve maternal health outcomes [12]. The HIV prevention and care continuum depends on numerous interconnected strategies, from testing, linkage to and retention in care, all of which are vital to the successful attainment of UN AIDS 90-90-90 goal [24]. Fundamentally, however, initial diagnosis of HIV is essential, and prior work has consistently emphasized the importance of universal testing [25]. Similarly, retention in care, particularly in resource-limited settings, creates numerous challenges both for individual patients and their communities [26, 27]. The village health worker program, specifically, may have played a role in the observed improvements. Prior work has demonstrated similar changes after the introduction of accompaniment models [28].

Finally, vaccine preventable infections contribute to high infant mortality worldwide [29, 30]. In Sub-Saharan Africa, there are numerous and complex barriers to immunization [31]. Extensive work has documented the economic value of vaccination programs [32]; one study concluded that an investment of $34 billion dollars for immunization programs in 94 low- and middle-income countries, would result in a savings of $586 billion dollars in reducing the cost of illness [33]. Thus, improving the rates of childhood immunization is of paramount importance. Our results are encouraging in that we observed an improvement in rates of complete 1-year immunizations. The specific barriers for local communities, however, will vary by region and must be the premise of future interventions.

While it is not possible to attribute those changes to any single intervention or constellation of interventions of the National Health Reform, prior work has demonstrated that the components selected have been associated with similar improvements. First, improving service delivery, from resource provision to assurance of the supply chain has been shown to improve healthcare utilization [34]. Second, decentralization of resources and personnel is a viral part of health systems strengthening, and is dependent upon the development of managerial capacity [35]. Improved health center management [36–38], and specifically improved district-level managerial capacity in resource-limited settings is associated with numerous improved health outcomes [39, 40]. Notably, concomitant assurance of accountability is essential to improved managerial capacity [41, 42], which in the present context was done via the community score card -itself a tool associated with demonstrable improvements in service provider effectiveness, service provision and accountability as well as responsiveness [43, 44]. Finally, the various roles and benefits of the community health worker model in resource-limited settings are well characterized [15, 45–47], and likely contributed significantly to the overall impact of the National Health Reform.

Importantly, as our results are observational in nature, direct attribution to the interventions of the National Health Reform are not appropriate. There are other potential explanations for at least some of the changes observed, such as evolving costs of medications and resources, government spending in other sectors that has indirect benefits on healthcare, and many more. Likewise, changes in demographics and risk-factors at the district level over time may have also confounded our results; however, in the absence of district-level data, the population level trends in demographics (Table 1) did not suggest significant difference before and after the interventions. Thus, further work is needed to delineate the causes for the improved metrics observed, which will not only facilitate more targeted interventions in the future but also facilitate replication of our results in other settings. However, in light of those findings, and previous work supporting similar models [28, 39, 40], we advocate for further support for infrastructure building and accompaniment models to continue progress towards improved maternal health outcomes. Infrastructure capacity building and assurance of medication supply chains, however, are equally valuable as the entire continuum is necessary for improvement in HIV care nationally.

## Future directions

As mentioned above, further assessment of the specific components of the intervention, while controlling for other potential explanations for improved health metrics, will be an essential next step. Additionally, review of the continuing barriers that prevent successful achievement of MDGs is warranted in order to further refine the National Health Reform. Further scale up of the health reform interventions into other districts and evaluation of similar outcomes is warranted, but simultaneous assessment of specific intervention components will be meaningful for iterative assessment of programmatic success. Similarly, future iterations of reform efforts should focus on improving retention in care, specifically antenatal care, which did not significantly change during our observation period. Further work is also needed to assess the impacts of the reform on Tuberculosis testing and care. Finally, one challenge we encountered was a paucity of district level census demographic data, making determination of the rate of change not possible. Such a challenge is likely to be evident in other similarly resource limited settings. Future efforts should strive for developing improved infrastructure for both population and disease surveillance, which may facilitate a more accurate understanding of population level rates of disease, as well as provide the foundation for public health responses to outbreaks.

## Conclusions

The Lesotho MOHSW and PIH developed and implemented a national health reform over the course of three years. We report the practice interventions deployed, highlighting three key areas: service delivery, health system management, and the village health worker program. We observed increases in maternal antenatal and post-natal care visits as well as emergency referrals, increases in child immunization rates, and improvements in retention in care among patients living with HIV as well as increased HIV testing. We also noted increased availability of essential health commodities. Although surrogates for concrete outcomes, those correlations are encouraging and support further research into the strategies employed with consideration of implementation of similarly multi-pronged health system strengthening interventions in other settings.

## Supporting information

**S1 Data. The table contains the raw, facility-level data, which was used to generate the results in the manuscript.**
(XLSX)

**S1 Text. This document is the 2019 annual report on technical support authored by Partners in Health regarding the pharmacy and medical supply chain management detailing augmentation of cold chain capacity for five health centers to appropriately store several life-saving medicines, as well as the interim outcomes for antenatal care visits, facility-based deliveries, and inter-facility transfer of complex cases.**
(PDF)

**S2 Text. This supplement provides additional details regarding the interventions done as a part of the national health reform.** It further includes a table demonstrating the three core areas identified as needs during the National Health Reform: Service Delivery, Health System Managerial Capacity, and the Professional Village Health Worker Program, and lists specific areas for targeted for improvement within each area. Those areas became the focus of the interventions within the National Health Reform.
(DOCX)

**S3 Text. This supplement provides the detailed study protocol.** Deviations from the protocol are highlighted in the methods section of the text.
(PDF)

**S4 Text. This supplement includes the standardized survey questions for key stakeholders as well as the data collection tool from which our outcome variables of interest were derived.**
(PDF)

**S1 Checklist. This supplement provides the STROBE checklist for reporting guidelines.**
(DOCX)

## Acknowledgments

The authors would like to acknowledge the government of Lesotho and leadership of Ministry of Health and Social Welfare. Further, we would like to acknowledge the Analysis Group Inc., for supporting the data analysis, and the Boston Consulting Group. Finally, the authors would like to acknowledge Pierre Y. Cremieux for his contributions to the work.

## Author Contributions

**Conceptualization:** Melino Ndayizigiye, Dan Schwarz, Joia S. Mukherjee.

**Data curation:** Lao-Tzu Allan-Blitz, Collin Whelley, Ermyas Birru, Ryan McBain, Di Miceli Andrea, Joia S. Mukherjee.

**Formal analysis:** Lao-Tzu Allan-Blitz, Di Miceli Andrea.

**Funding acquisition:** Dan Schwarz, Joia S. Mukherjee.

**Investigation:** Seyfu Abebe, Afom Andom, Emily Gingras, Mathabang Mokoena, Meba Msuya, Patrick Nkundanyirazo, Thiane Mohlouoa, Joalane Mabathoana, Palesa Chetane, Likhapha Ntlamelle, Ermyas Birru, Ryan McBain.

**Methodology:** Melino Ndayizigiye, Emily Dally, Afom Andom, Retsepile Tlali, Emily Gingras, Mathabang Mokoena, Meba Msuya, Thiane Mohlouoa, Fusi Mosebo, Sophie Motsamai, Palesa Chetane, Likhapha Ntlamelle, Joel Curtain, Collin Whelley, Ermyas Birru, Ryan McBain, Joia S. Mukherjee.

**Project administration:** Emily Dally, Seyfu Abebe, Retsepile Tlali, Emily Gingras, Mathabang Mokoena, Meba Msuya, Patrick Nkundanyirazo, Fusi Mosebo, Sophie Motsamai, Joalane Mabathoana, Likhapha Ntlamelle, Joel Curtain, Collin Whelley, Joia S. Mukherjee.

**Resources:** Emily Dally, Retsepile Tlali, Mathabang Mokoena, Patrick Nkundanyirazo, Thiane Mohlouoa, Palesa Chetane, Collin Whelley, Ermyas Birru.

**Supervision:** Melino Ndayizigiye, Afom Andom, Retsepile Tlali, Thiane Mohlouoa, Fusi Mosebo, Sophie Motsamai, Joalane Mabathoana, Joel Curtain, Ermyas Birru, Dan Schwarz, Joia S. Mukherjee.

**Writing – original draft:** Lao-Tzu Allan-Blitz.

**Writing – review & editing:** Melino Ndayizigiye, Lao-Tzu Allan-Blitz, Emily Dally, Seyfu Abebe, Afom Andom, Retsepile Tlali, Emily Gingras, Mathabang Mokoena, Meba Msuya, Patrick Nkundanyirazo, Thiane Mohlouoa, Fusi Mosebo, Sophie Motsamai, Joalane Mabathoana, Palesa Chetane, Likhapha Ntlamelle, Joel Curtain, Collin Whelley, Ermyas Birru, Ryan McBain, Di Miceli Andrea, Dan Schwarz, Joia S. Mukherjee.

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
