## [Decision Letter · Decision Letter 0]

9 May 2022

PGPH-D-22-00040

Improving Access to Health Services Through Health Reform in Lesotho: Progress Made Towards Achieving Universal Health Coverage

Dear Dr. Lao-Tzu Allan-Blitz

Thank you for submitting your manuscript to PLOS Global Public Health. After careful consideration, we feel that it has merit but does not fully meet PLOS Global Public Health’s publication criteria as it currently stands. Therefore, we invite you to submit a revised version of the manuscript that addresses the points raised during the review process.

I agree with the reviewers that there is need to add more information in the methods section and be explicit on what was done and that should trickle to the presentation of the results which will also have a bearing on the discussion. Once this is clarified, it will highlight the assertions you are making in the conclusion.

We look forward to receiving your revised manuscript.

Kind regards,

Alinane Linda Nyondo-Mipando, PhD

Academic Editor

JOURNAL REQUIREMENTS:

2. In the online submission form, you indicated that "Data are available upon request". All PLOS journals now require all data underlying the findings described in their manuscript to be freely available to other researchers, either 1. In a public repository, 2. Within the manuscript itself, or 3. Uploaded as supplementary information.

3. We noticed that you used “data not shown”/"unpublished data" in the manuscript. We do not allow these references, as the PLOS data access policy requires that all data be either published with the manuscript or made available in a publicly accessible database. Please amend the supplementary material to include the referenced data or remove the references.

4. All figures and supporting information files will be published under the Creative Commons Attribution License (creativecommons.org/licenses/by/4.0/). Authors retain ownership of the copyright for their article and are responsible for third-party content used in the article. 

Figure 1: please (a) provide a direct link to the base layer of the map used and ensure this is also included in the figure legend; (b) provide a link to the terms of use / license information for the base layer. We cannot publish proprietary or copyrighted maps (e.g. Google Maps, Mapquest) and the terms of use for your map base layer must be compatible with our CC-BY 4.0 license. 

Please upload any written confirmation as an 'Other' file type. It must clarify that the copyright holder understands and agrees to the terms of the CC BY 4.0 license; general permission forms that do not specify permission to publish under the CC BY 4.0 will not be accepted. Note that uploading an email confirmation is acceptable.

5. Please provide separate figure files in .tif or .eps format. Kindly, removed them from the manuscript file.

6. We have noticed that you have uploaded Supporting Information files, but you have not included a list of legends. Please add a full list of legends for your Supporting Information files after the references list. 

Reviewers' comments:

Reviewer's Responses to Questions

**Comments to the Author**

1. Does this manuscript meet PLOS Global Public Health’s publication criteria? Is the manuscript technically sound, and do the data support the conclusions? The manuscript must describe methodologically and ethically rigorous research with conclusions that are appropriately drawn based on the data presented.

Reviewer #1: No

Reviewer #2: No

2. Has the statistical analysis been performed appropriately and rigorously?

Reviewer #1: I don't know

Reviewer #2: No

3. Have the authors made all data underlying the findings in their manuscript fully available (please refer to the Data Availability Statement at the start of the manuscript PDF file)?

Reviewer #1: No

Reviewer #2: No

4. Is the manuscript presented in an intelligible fashion and written in standard English?

Reviewer #1: Yes

Reviewer #2: Yes

5. Review Comments to the Author

Reviewer #1: Thank you for this opportunity to provide a peer review for this interesting manuscript titled “Improving Access to Health Services through Health Reform in Lesotho: progress made towards achieving Universal Health Coverage.”

The study basically set out to assess changes in quality metrics from Pre- to Post-National Health Reform during the period of 2013 to 2017 in the Kingdom of Lesotho. A baseline assessment was done in 70 facilities in four districts of Lesotho (making up catchment area of about 40% of the country’s population). The study observed improvements in Maternal health outcomes, HIV diagnostic capacity, and Childhood Immunization Rates. This study makes a case for encouraging and supporting further research into strategies utilized for implementation of multi-faceted health system strengthening interventions especially in developing countries.

The fact that the study aims to assess the impact of Health Reforms in a developing country; an area where there is paucity of data is something to be encouraged. There are however areas that the Authors need to clarify in my opinion.

1. In the Methods Section, three areas were referred to as having deficits; namely supply-chain, district managerial capacity and Village Health Worker program. My observation is that there are no results to show if there was any difference in these areas after the National Health Reforms were introduced.

2. The authors may wish to explain how they were able to arrive at the number of Health Facilities to evaluate, and how they came about the specific facilities and the Regions in Lesotho.

3. The authors may wish to mention measures taken to ensure comparability between the results of the two assessments (Pre and Post National Health Reform).

4. The conclusion by the authors that there was “improvement in health outcomes among Maternal and Child health, as well as HIV testing and availability of essential health commodities” may be difficult to justify, as essentially what the study reported was an increase in utilization rates.

Once again, I appreciate the opportunity given me.

Kind Regards.

Reviewer #2: Provide study data

What was the study design? Was this a quasi-experimental study? (A controlled or uncontrolled before-and-after study)? If so, please state the duration of the reform period and at what point the endline assessment was done.

Study results in abstract are reported as absolute numbers yet project aimed to improve metrics that are reported in rates (to cater for population size changes). Please revise e.g. instead of reporting on number of new ANC attendances, you could report on proportion of expected new ANC attendance per number of estimated pregnancies; or for immunisation, immunisation coverage instead of absolute numbers.

6. PLOS authors have the option to publish the peer review history of their article (what does this mean?). If published, this will include your full peer review and any attached files.

**Do you want your identity to be public for this peer review?** For information about this choice, including consent withdrawal, please see our Privacy Policy.

Reviewer #1: No

Reviewer #2: No

---

## [Decision Letter · Decision Letter 1]

18 Aug 2022

PGPH-D-22-00040R1

Improving Access to Health Services Through Health Reform in Lesotho: Progress Made Towards Achieving Universal Health Coverage

Dear Dr. Lao-Tzu Allan-Blitz

Thank you for submitting your manuscript to PLOS Global Public Health. After careful consideration, we feel that it has merit but does not fully meet PLOS Global Public Health’s publication criteria as it currently stands. Therefore, we invite you to submit a revised version of the manuscript that addresses the points raised during the review process.

Your report is interesting and includes a detailed description of what sounds like a very important set of health system strengthening activities. As it currently stands it is not written adequately as an Original Research article and will require very substantial changes to meet our publication standards. If you do not wish to make such substantive changes, I would suggest writing it up as a Field Report (for a different journal), rather than original research, as this will allow you to focus on description of intervention and present general results without expecting the scientific rigour of original research (but it would need to be much shorter).

If you wish to pursue publication as Original Research in PLoS GPH, please review again the comments by the previous Editor and Reviewers, and my additional comments here. I acknowledge that you have partially addressed some of the previous comments but there are still major deficiencies in your reporting, particularly of Methods. I will try and be frank and clear about what these issue are and what you need to do about them.

Please refer to a reporting standard, such as STROBE or TREND. Include the completed reporting standard checklist with your revised submission. Pay particular attention to the reporting elements needed in the Methods section. You have provided a very long description of the development and content of the intended intervention - this can be heavily cut (with details in supplement). You have provided almost no information on the outcomes, data collection, or analysis. For example, what were the primary/secondary outcomes, why did you select these (and not others). You say you did 'time series' analysis, but provide no details - and the results do not present any time series analysis results. Please address each of the essential reporting elements in full.The Introduction can be heavily cut. Please include a clear Objective and/or Hypothesis that you seek to test.Please include the study protocol as a supplemental file, and data analysis plan if available. And make clear in the reporting where you deviated from the pre-specified protocol (including how the intervention evolved over time, and differences in the data you collected, how you analysed it and reported it). Most health systems interventions will not have been delivered (or evaluated) exactly as planned - and this is important information to disclose.Your authorship list is overwhelmingly US-centric. You cite involvement of the MOHSW in planning, implementation, and data collection. I find it concerning that you did not involve MOHSW in authorship, both from a research validity perspective (they will understand things you do not) and from a research impact perspective (they will be prime users of this data). Please provide explanation in your Reflexivity statement about why you included only a single author from Lesotho and no government partners. 

Please submit your revised manuscript by 04 August 2022.  If you will need more time than this to complete your revisions, please reply to this message or contact the journal office at globalpubhealth@plos.org. Please include the following items when submitting your revised manuscript:

We look forward to receiving your revised manuscript.

Kind regards,

Hamish Graham, MBBS, MPH, MSSc, PhD

Academic Editor

Journal Requirements:

1. "Allan-Blitz - Figure 1 - 5.25.22.tiff", "Allan-Blitz - Figure 2 - 5.25.22.tiff", "Allan-Blitz - Figure 3 - 5.25.22.tiff", and "Allan-Blitz - Figure 4 - 5.25.22.tiff" files are over our file size limit of 10MB. Please reduce the file size to no more than 10MB. For further help on compressing figures visit: https://journals.plos.org/globalpublichealth/s/figures

2. Please include your main table (Table 1) as part of your main manuscript and remove the individual file. Please note that supplementary tables should remain as separate "Supporting Information" files.

3. We noticed that you used “unpublished”/“unpublished data” in the manuscript. We do not allow these references, as the PLOS data access policy requires that all data be either published with the manuscript or made available in a publicly accessible database. Please amend the supplementary material to include the referenced data or remove the references.

4. Please update your online Competing Interests statement. If you have no competing interests to declare, please state: “The authors have declared that no competing interests exist.”

5. Please amend your online detailed Financial Disclosure statement. This is published with the article. It must therefore be completed in full sentences and contain the exact wording you wish to be published.
---

## [Editor Report · Decision Letter 2]

11 Oct 2022

Improving Access to Health Services Through Health Reform in Lesotho: Progress Made Towards Achieving Universal Health Coverage

PGPH-D-22-00040R2

Dear Dr. Allan-Blitz,

We are pleased to inform you that your manuscript 'Improving Access to Health Services Through Health Reform in Lesotho: Progress Made Towards Achieving Universal Health Coverage' has been provisionally accepted for publication in PLOS Global Public Health.

Best regards,

Hamish R Graham

Academic Editor